# Urinary MicroRNAs as Biomarkers of Urological Cancers: A Systematic Review

**DOI:** 10.3390/ijms241310846

**Published:** 2023-06-29

**Authors:** Achille Aveta, Simone Cilio, Roberto Contieri, Gianluca Spena, Luigi Napolitano, Celeste Manfredi, Antonio Franco, Fabio Crocerossa, Clara Cerrato, Matteo Ferro, Francesco Del Giudice, Paolo Verze, Francesco Lasorsa, Andrea Salonia, Rajesh Nair, Jochen Walz, Giuseppe Lucarelli, Savio Domenico Pandolfo

**Affiliations:** 1Department of Neurosciences, Reproductive Sciences and Odontostomatology, Urology Unit, University of Naples “Federico II”, 80138 Naples, Italy; achille-aveta@hotmail.it (A.A.); simocilio.av@gmail.com (S.C.); spena.dr@gmail.com (G.S.); luigi.napolitano@unina.it (L.N.); 2Department of Urology, Institut Paoli-Calmettes Cancer Centre, 13055 Marseille, France; walzj@ipc.unicancer.fr; 3Division of Experimental Oncology/Unit of Urology, URI, IRCCS Ospedale San Raffaele, 20132 Milan, Italy; salonia.andrea@hsr.it; 4Department of Biomedical Sciences, Humanitas University, 20072 Pieve Emanuele, Italy; roberto.contieri@humanitas.it; 5Unit of Urology, Department of Woman, Child and General and Specialized Surgery, University of Campania “Luigi Vanvitelli”, 80131 Naples, Italy; manfredi.celeste@gmail.com; 6Department of Urology, Sant’Andrea Hospital, “La Sapienza” University, 00189 Rome, Italy; anto.franco@hotmail.it; 7Department of Urology, Magna Graecia University of Catanzaro, 88100 Catanzaro, Italy; crocerossa@unicz.it; 8Urology Unit, University Hospital Southampton NHS Trust, Southampton SO16 6YD, UK; clara.cerrato01@gmail.com; 9Division of Urology, European Institute of Oncology, 20122 Milan, Italy; matteo.ferro@ieo.it; 10Department of Urology, University Sapienza, 00185 Rome, Italy; francesco.delgiudice@uniroma1.it; 11Department of Medicine and Surgery, Scuola Medica Salernitana, University of Salerno, 84081 Fisciano, Italy; pverze@unisa.it; 12Urology, Andrology and Kidney Transplantation Unit, Department of Precision and Regenerative Medicine and Ionian Area, University of Bari “Aldo Moro”, 70124 Bari, Italy; francesco-lasorsa96@libero.it (F.L.); giuseppe.lucarelli@uniba.it (G.L.); 13The Urology Centre, Guy’s and St. Thomas’ NHS Foundation Trust, Guy’s Hospital, London SE1 9RT, UK; drrajnair@hotmail.com

**Keywords:** microRNAs, biomarkers, urological cancers, prostate cancer, bladder cancer, upper tract urothelial carcinoma, renal cancer

## Abstract

MicroRNAs (miRNAs) are emerging as biomarkers for the detection and prognosis of cancers due to their inherent stability and resilience. To summarize the evidence regarding the role of urinary miRNAs (umiRNAs) in the detection, prognosis, and therapy of genitourinary cancers, we performed a systematic review of the most important scientific databases using the following keywords: (urinary miRNA) AND (prostate cancer); (urinary miRNA) AND (bladder cancer); (urinary miRNA) AND (renal cancer); (urinary miRNA) AND (testicular cancer); (urinary miRNA) AND (urothelial cancer). Of all, 1364 articles were screened. Only original studies in the English language on human specimens were considered for inclusion in our systematic review. Thus, a convenient sample of 60 original articles was identified. UmiRNAs are up- or downregulated in prostate cancer and may serve as potential non-invasive molecular biomarkers. Several umiRNAs have been identified as diagnostic biomarkers of urothelial carcinoma and bladder cancer (BC), allowing us to discriminate malignant from nonmalignant forms of hematuria. UmiRNAs could serve as therapeutic targets or recurrence markers of non-muscle-invasive BC and could predict the aggressivity and prognosis of muscle-invasive BC. In renal cell carcinoma, miRNAs have been identified as predictors of tumor detection, aggressiveness, and progression to metastasis. UmiRNAs could play an important role in the diagnosis, prognosis, and therapy of urological cancers.

## 1. Introduction

Genitourinary cancers can affect the kidney, upper urinary tract, bladder, prostate, testis, and penis. In developed countries, kidney, bladder, and prostate cancers are the three major types of genitourinary cancers, with 166,440, 228,730, and 486,840 worldwide deaths in 2019, respectively [1]. Because of the associated high morbidity and mortality, the improvement in early diagnosis is crucial for better clinical results. Although the identification of emerging targets and novel molecules has resulted in encouraging progress in the management of genitourinary tumors, valuable tools for cancer diagnosis and follow-up continue to be lacking. Therefore, the research for reliable prognostic and predictive biomarkers for the early diagnosis of patients with genitourinary malignancies is actually an evolving landscape. In this context, an ideal biomarker should ensure high-accuracy results and should be as minimally invasive as possible to obtain [2].

In the last few years, microRNAs (miRNAs) emerged as useful markers thanks to their occurrence in all tissues. Indeed, normal and cancerous cells can use exosomes to secrete these molecules into blood or urine as free-circulating miRNAs [3]. An increasing number of studies also suggest that miRNAs have great promise to serve as novel biomarkers in liquid biopsy [4].

MiRNAs are 20–25-nucleotide-long noncoding single-stranded RNA molecules which regulate gene expression, through the breakdown of the mRNA transcript or the inhibition of the translation of the mRNA to protein [5]. 

Changes in the expression of miRNAs have been associated with the progression of different cancers [6]. Indeed, miRNAs can disturb the expression of oncogenic or tumor-suppressive target genes implicated in cancer pathogenesis [7]. Notably, several miRNAs have been found to be upregulated or downregulated in various tumors, with an oncogenic or oncosuppressive role [8]. 

MiRNAs are emerging as diagnostic tools in several tumors. Previous studies assessed their levels in surgical and liquid samples from patients with cancers. However, samples from operative specimens could be altered by coagulation, necrosis, and formalin fixation. In this context, urinary miRNAs (umiRNAs) gain advantages from bypassing alternating processes and form the reduced vulnerability to urinary RNase in urine [9,10]. Furthermore, compared to a local tumor sample, urine is a readily accessible source that does not need invasive procedures and represents the genetic profile of the entire tumor [11,12].

In consequence, considering the high worldwide prevalence of genitourinary cancers and the growing interest in the role of miRNAs as non-invasive diagnostic biomarkers, the current systematic review aims at summarizing the role of umiRNAs in all genitourinary cancers in order to lay the foundation for further validation studies.

## 2. Methods

A systematic review of the literature was performed in March 2023 using the PubMed^®^, Scopus^®^, Web of Science^®^, Clinicaltrial.gov, and Cochrane Library^®^ databases (MEDLINE, EMBASE, and Web of Science databases). Preferred Reporting Items for Systematic Review and Meta-Analysis (PRISMA) recommendations were followed to design the search strategies, selection criteria, and evidence report. The International Prospective Register of Systematic Review (PROSPERO) protocol number is CRD42023402737. Patient-related and intervention search terms were combined to build the following search string: (urinary miRNA) AND (prostate cancer); (urinary miRNA) AND (bladder cancer); (urinary miRNA) AND (renal cancer); (urinary miRNA) AND (testicular cancer); (urinary miRNA) AND (urothelial cancer).

Search results were filtered by language (English only), species (human), and publication type (article). Study eligibility was defined using the PICOS (patient, intervention, comparator, outcome, study type) approach. Inclusion criteria were:

(P) studies focused on adults (>18 yrs of age) with a diagnosis of kidney, bladder, or prostate cancers.

(I) identification of miRNAs as diagnostic biomarkers.

(C) in which controls as healthy subjects were used as a comparator.

(O) evaluating one or more of the following outcomes: in the diagnosis, prognosis, and therapy of urological cancers.

(S) retrospective or prospective comparative studies, with a minimum cohort size of 10 patients.

Exclusion criteria were: (1) SP on animal or cadaveric models; (2) studies reporting fewer than five cases; and (3) non-original studies, including editorial comments, meeting abstracts, case reports, or letters to the editor, or any form of grey literature, because of the general lack of details or peer review.

## 3. Results and Discussion

### 3.1. Literature Search Results

The PRISMA flow chart of the study selection process is shown in Figure 1. An initial search identified 1364 studies. Of these, 253 were excluded due to duplication. After applying selection criteria, another 989 records were excluded. A total of 36 studies, including over 3900 patients, were included in the systematic review. Sixteen studies, including 2498 patients, reported data for bladder cancer (Table 1). Ten studies, including 641 patients, reported data on miRNA for prostate cancer (Table 2). Eight studies, including 521 patients, reported data on renal cancer. (Table 3). Two studies with an overall total of 240 patients reported results for urothelial cancer (Table 4). 

### 3.2. Results According to Main Topic

#### 3.2.1. The Role of MicroRNA in the Detection of Bladder Cancer

Bladder cancer (BC) is one of the most common urogenital cancers worldwide [48]. Among the other types, urothelial carcinoma (UC) is the most common histological type, with a prevalence of almost 90% [49]. According to European Association of Urology (EAU) guidelines, the diagnosis of BC is actually based on imaging, cytology, cystoscopy, and histopathological analysis of sampled tissue from either cold-cup biopsy or trans-urethral resection (TURB) [50,51]. Based on histological manifestations and biological traits, BC can be classified in non-muscle-invasive bladder cancer (NMIBC) and muscle-invasive bladder cancer (MIBC) [52].

Although many biomarkers have been evaluated as potential diagnostic tools of BC, none of them have reached adequate accuracy to replace cystoscopy and cytology. However, being that these diagnostic methods are uncomfortable for the patients and expensive for the healthcare system, new biomarkers such as umiRNAs are crucial when BC is suspected [53,54].

Since the 2010s, numerous studies have demonstrated that different types and concentrations of umiRNAs could predict the carcinogenicity and invasiveness of BC. Our systematic report found a large number of miRNAs or panels with variable expressions and different predictive features, facilitating the diagnosis, prognosis, and recurrence of BC. Notably, some umiRNAs have the potential to be valid markers in BC detection and the differentiation of NMIBC and MIBC. On the other hand, other umiRNAs have predictable power in discriminating patients with NMIBC from patients with cystitis or with nonmalignant hematuria. Finally, a statistical association has been reported, with shorter recurrence-free survival times as a proxy for NMIBC recurrence.

For instance, Hanke et al. found that urine samples of BC patients contained increased levels of miR-126, miR-182, and miR-199a. In addition, the authors reported that a higher RNA ratio (miR 126: miR 152) could enable the detection of BC with a specificity of 82% and sensitivity of 72% (AUC= 0.768) [29]. Moreover, Kim et al. showed overexpression of miR-214 in the urine samples of NMIBC patients compared to control specimens (20.08 ± 3.21 vs. 18.96 ± 2.68, *p* = 0.002) [26].

Furthermore, lower levels of miR-214 were associated with a significantly longer recurrence-free survival time, making it an independent predictor of NMIBC recurrence (*p* = 0.012) [26].

Overexpression of miR-155 in NMIBC patients has been reported by Zhang et al. The authors showed how the tested umiRNA allows discrimination between patients with NMIBC from patients with cystitis and healthy controls, with 80.2% sensitivity and 84.6% specificity [25].

Nevertheless, the relationship between miR-155 overexpression and bladder cancer development has not been fully elucidated. A possible explanation is the action of miR-155 in promoting some tumor cell growth via Wnt/β-catenin signaling activation ([55], p. 155).

In 2018, Piao et al. explored a novel method to discriminate bladder cancer from benign hematuria by measuring the urinary miR-6124 to miR-4511 ratio. The capacity of this proposed diagnostic tool enabled the discrimination of BC from patients with hematuria under nonmalignant conditions, with a sensitivity higher than 90% (AUC: 0.888, 91.5% SN, 76.2% SP) (*p* < 0.001) [21].

Differences in miR-20a expression in the urine samples from 80 NMIBC patients and 86 healthy individuals were investigated by Huang et al. They found that urinary concentrations of miR-20a were significantly higher in NMIBC patients than in healthy controls (*p* < 0.001). Moreover, they showed that a larger tumor size and advanced tumor grade were associated with a high expression of this umiRNA (all *p* < 0.05) [19].

Sasaki et al. demonstrated that the expression level of miR-146a-5p in patients with BC was higher than in healthy individuals (AUC = 0.773, 95% CI, 0.701–0.892) (*p* = 0.014). Higher umiR-146a-5p concentrations were displayed in patients with high-grade BC and with MIBC with respect to those with low-grade tumors (*p* = 0.0436) or NMIBC (*p* = 0.1391). Moreover, the authors showed that levels of miR-146a-5p decreased to the normal range after TURB [24].

In the multitude of studies on BC, umiR-146 showed the most overlap. In this context, Andreu et al. reported overexpression in low-grade rather than in high-grade disease, whereas Baumgart et al. found overabundance in high-grade more than in lower-graded disease [16,23]. Nevertheless, both the research groups showed that this umiR-146 in BC is indeed an inflammasome, and the discordance of results might be explained by its inflammatory status in BC and not directly by the aggressiveness of the disease; thus, further studies are needed to clarify its role.

Therefore, in addition to the overexpression of umiRNAs in BC, several studies have shown the downregulation of tumor suppressor miRNAs in urinary samples. In 2012, Yun et al. reported the downregulation of miR-145 in NMIBC and MIBC patients compared to healthy controls (77.8% sensitivity and 61.1% specificity for NMIBC, AUC 0.729; 84.1 and 61.1% for MIBC, respectively, AUC 0.790, *p* < 0.001). Moreover, miR-145 urinary levels were lower in MIBC patients than in NMIBC patients (*p* = 0.036). In addition, in the same study, the authors reported that the levels of miR-200a were also significantly decreased in NMIBC and MIBC compared to healthy controls (*p* < 0.001) [28].

In contrast to the previous finding, in 2023, Mamdouh et al. reported that the urinary concentrations of miR-200, miR-145, and miR-21 were higher in cases of low- and high-grade BC compared to the controls, depicting a possible oncogenic role of those miRNAs (*p* = 0.02, 0.01 and 0.05, respectively) [13].

To improve the accuracy of using umiRNAs for the detection of BC, numerous studies analyzed combination tests utilizing multiple umiRNAs. For instance, Mengual et al. identified a subset of six umiRNAs (miR-187, miR-18a*, miR-25, miR-142-3p, miR-140-5p, and miR-204), establishing a specificity of 86.5% and a sensibility of 84.8% (AUC 0.92) in the diagnosis of BC [27].

Three urine microRNAs, miR-21-5p, miR-141-3p, and miR-205-5p, have been found by Ghorbanmehr et al. to be prospective non-invasive diagnostic biomarker candidates for the identification of both bladder and prostate cancer [20].

Hofbauer et al. achieved comparable results with 88.3% sensitivity using six different umiRNAs (let-7c, miR-135a, miR-135b, miR-148a, miR-204, and miR-345), which can predict the presence of BC from urine samples, independently from grading and staging (AUC 0.88) [22].

Pardini et al. also confirmed, by firstly using Next-Generation Sequencing (NGS), that the combination of specific miRNA profiles may provide more robust results than individual miRNAs. Indeed, the authors showed a statistically significant improvement in the AUC discrimination between BC and controls (from 50% to 70%), using a set of three umiRNAs (miR-30a-5p, let-7c-5p, and miR-486-5p) [18]. Accordingly, Braicu et al. proposed interactions between the genes associated with BC carcinogenesis (TP53, FGFR3, KDR, PIK3CA, and ATM) and altered miRNAs’ expressions (miR-139-5p, miR-143-5p, miR-23a-3p, miR-141-3p, miR-205-5p). In particular, three upregulated miRNAs (miR-141b, miR-200 s, and miR-205) and two downregulated (miR-139-5p and miR-143-5p) target these multiple genes involved in the carcinogenesis of bladder cancer [17].

Likewise, Lin et al. in 2021 concluded that let-7b-5p, miR-149-5p, miR-146a-5p, miR-193a-5p, and miR-423-5p were significantly increased in BC compared with healthy specimens. Moreover, these umiRNAs had a significant impact on cancer-related signaling pathways implied in cell growth, proliferation, and survival, such as: PI3K/AKT, MAPK, focal adhesion, and Erb [15,56].

In 2022, Moisoiu et al. firstly demonstrated an AUC of 0.92  ±  0.06 in discriminating patients with BC from controls using the combination of surface-enhanced Raman spectroscopy (SERS) with three differentially expressed miRNAs (miR-34a-5p, miR-205-3p, miR-210-3p). This unique method seems to guarantee a better BC diagnostic and molecular stratification, even if studies in larger cohorts should be performed to confirm these results [14].

In conclusion, in relation to the T stage, De Long et al. identified seven miRNAs overexpressed in the bladder cancer group (*p* < 0.05). Of the RNA analyzed, miR-940 was differentially expressed between patients with MIBC compared with patients with NMIBC. In particular, miR-940 levels were the highest in advanced disease (pT1 G3 and ≥pT2) and the lowest in the absence of tumor (healthy volunteers with or without history of urothelial carcinoma) [57]. Contrarily, Baumgart et al. demonstrated a downregulation of miR-138-5p between pT2 and pT3–4 tumors, indicating that low expression correlates with an aggressive phenotype [16]. This might be explained by the fact that low expression of this miRNA, as shown in another two studies, results in higher expression of the EMT-associated protein ZEB2 [58,59].

#### 3.2.2. The Role of MicroRNA in the Detection of Prostate Cancer

Prostate cancer (PCa) is the most commonly diagnosed malignancy among men and the fifth cause of male cancer-related death worldwide [49,60]. The suspicion of prostate cancer arises from an abnormal digital rectal examination or/and an elevated PSA value [61]. The gold standard for the diagnosis is then obtained by transperineal or transrectal prostate biopsy [62,63]. Although serum prostate-specific antigen (PSA) is the most widely used biomarker for prostate cancer (PCa) screening, it has several limitations. The lack of specificity and the limited ability of this serum marker to distinguish between malignant and benign causes of its elevation might in fact result in overdiagnosis and a significant risk of false positive results [64].

To overcome these limitations, numerous studies have been conducted to identify new biomarkers, and several miRNAs have been shown to be involved in the development and progression of PCa [65,66,67]. In the selected studies, we found that umiRNAs could be used as valuable tools in differentiating PCa from benign rising of PSA. Some other selected studies showed the role of umiRNAs in predicting prognosis and progression.

The first studies investigated the miRNAs’ profile directly in prostate carcinoma tissue. Indeed, in 2009, Schaefer et al. firstly showed the upregulation of miR-183 and the downregulation of miR-205 in PCa tissues [68].

After that, in 2015, Stephan et al. aimed to translate these results into a urine-based testing procedure. They enrolled 38 patients with PCa and 38 without PCa to test the clinical utility of miR-183 and miR-205 in urines samples, and found that urinary concentration of those miRNAs were comparable in patients with and without PCa [38].

Salido-Guadarrama et al. showed that elevated urinary levels of miR-100 and miR-200b were associated with advanced PCa [37]. Furthermore, miR-100 remained upregulated throughout the carcinogenic process, and its downregulation has been observed for hormone-refractory PCa [69,70].

In 2017, Rodriguez et al. showed that miR-196a-5p and miR-501-3p, downregulated in urinary exosomes, are promising biomarkers for PCa [36]. In the same year, Foj et al. observed that, when compared to samples from healthy men, the urinary pellet of PCa patients had higher concentrations of miR-21, miR-141, and miR-375 (*p* 0.001, 0.033, and 0.038, respectively). On the other hand, based on the study by Nadiminty et al. [71], they found no significant differences in the expression of let-7c. Moreover, they also found a higher expression of miR-141 in patients with higher Gleason scores (*p* = 0.034) [35].

Supporting these notions, Ghorbanmehr et al. collected urines samples from 110 men with BC (*n* = 45), PCa (*n* = 23) cancer, and benign prostatic hyperplasia (BPH) (*n* = 22), and from healthy men (*n* = 20). They assessed the expression of miR-21-5p, mi-R-141-3p, and miR-205p to identify and discriminate PCa patients from those with BPH (p 0.001, 0.005, and 0.020, respectively). Moreover, the authors reported how the upregulation of those miRNAs in urine samples was associated with higher cancer detection specificity in PCa compared to PSA testing [20].

Markert et al. analyzed urine samples of 53 patients (25 with BPH and 28 with PCa) and showed that miR-19b and miR-26a were significantly downregulated in PCa patients compared to BPH patients [34]. These microRNAs seem to play a role in regulating PTEN (phosphatase and tensin homolog enzyme), whose mutation is a common event in Pca [72].

In 2021, Hasanoglu et al. identified miR-320a as a valuable biomarker in the diagnosis of PCa, reporting higher concentrations in PCa patients compared to healthy controls (*p* = 0.0168) [33]. The upregulation of this microRNA confirmed what Porkka et al. reported in a previous study [70].

Over the years, ratio analysis has been used to improve results in microRNA research. It consists of measuring and comparing the expression ratios of upregulated to downregulated miRNAs in PCa and control patients. Using this approach, Byun et al. observed that the urinary miR-1913 to miR-3659 ratio was increased in PCa (AUC = 0.7, 95% CI, 61.4% SN, 71.8% SP), declaring a particular utility in patients within the PSA grey zone (defined as total serum PSA between 3 to 10 ng/mL) [32].

In addition, in Kang and colleagues’ study, the expression ratio of urinary miR-H9 to miR-3659 was quantified, and they affirmed that the ratio was significantly higher in the PCa group than in the healthy men group (AUC = 0.803, 95% CI) (*p* < 0.001), and that it could represent a non-invasive biomarker for PCa [31].

In conclusion, umiRNAs could serve as a biomarker supplemental to PSA for the diagnosis, but also for the prediction, of cancer progression, according to the latest studies. Indeed, in 2022, Lee et al. reported that miR-21-5p, miR-574-3p, and miR6880-5p were significantly higher in patients with CRPC (castration-resistant prostate cancer) and they could be used as potential biomarkers for the prognosis of CRPC [30]. In particular, overexpression of miR-21-5p downregulated programmed cell death protein 4, which is a regulator of PCa cell growth and castration resistance, whereas the overexpression of miR-574-3p reflected the downregulation of the Notch signaling pathway, DNA damage, and apoptosis [73,74].

#### 3.2.3. The Role of MicroRNA in the Detection of Renal Cancer

Renal cell carcinoma (RCC) is the 6th most frequently diagnosed cancer in men and the 10th in women, representing the 3rd most frequent genitourinary malignancy worldwide and the 13th most common cause of cancer death worldwide [75,76,77,78,79,80,81,82].

Symptoms related to RCC are usually rare, and occur in the late stages [83,84,85]. In this context, several microRNAs have been tested and identified as early diagnostic markers or as useful tools in the follow-up of treated patients [86]. Overall, Cui and Cui observed a significant positive correlation between human tissue miRNAs and the ones from urine specimens in patients with renal cancer (rho = 0.51, *p* < 0.001) [87].

Findings from our systematic review show that increasing levels of several umiRNAs and panels are related to a higher probability of diagnosing malignant renal masses, whereas other umiRNAs could be helpful in differentiating benign masses, such as oncocytoma.

In 2012, von Brandenstein et al. enrolled 25 patients with ccRCC and 5 healthy volunteers. They found that miR-15a levels from paraffin-embedded tissue and from urine samples are inversely related in malignant versus benign renal tumors. Thus, the authors suggested miR-15a as a potential new preoperative urinary marker for patients with renal cancer [88].

Fedorko et al. analyzed the role of the miRNA let-7 family, which is widely accepted as a tumor suppressor miRNA. Indeed, downregulation of the members of the let-7 family has been observed in various types of tumor tissue, including RCC, whereas its upregulation has been observed in BC [89,90]. For the specific purpose of their study, the authors analyzed urine samples of 69 patients with non-metastatic ccRCC and 36 healthy controls. They identified six let-7 miRNAs (let-7 let-7a, let-7b, let-7c, let-7d, let-7e, and let-7g) highly expressed in the urine of ccRCC patients with respect to healthy controls (all *p* < 0.015); in particular, let-7a outperformed the others, and may be considered a promising non-invasive biomarker for the detection of clear-cell RCC [45].

Li et al. collected urinary samples from 75 patients diagnosed with ccRCC, 45 healthy volunteers, and, to determine a decrease in umiRNAs’ concentration after surgery, they repeated the collection of urinary samples in 15 patients 7 days after tumor resection. The authors identified that free miR-210 levels were significantly higher in patients with ccRCC than in control subjects (*p* < 0.001), regardless of tumor staging. Moreover, miR-210 levels were significantly reduced one week after surgery, thus directly reflecting the presence of ccRCC [44].

In 2018, Mytsyk et al. aimed at testing the utility of urinary miR-15a as a diagnostic molecular biomarker of ccRCC. They collected urinary samples from 67 patients with various solid renal tumors and 15 healthy controls. MirR-15a allowed them to discriminate between malignant and benign renal masses (*p* < 0.01), and its levels were significantly reduced after one week from tumor surgery. Thus, the authors affirmed that mir-15a could be used as a reliable marker for the diagnosis of ccRCC [43].

Song et al. detected the expression of dysregulated miRNAs in urine exosomes of ccRCC patients and healthy individuals, in order to identify a specific dysregulated miRNA. They identified several umiRNAs in patients with ccRCC, PCa, and BC, and healthy individuals. Among them, the expression levels of miR-30c-5p in the urinary exosomes of ccRCC patients were significantly lower than that of normal individuals. The sensitivity and specificity of urinary exosome miR-30c-5p in the diagnosis of ccRCC were found to be 68.57% and 100%, respectively [41].

In 2020, Cochetti et al. identified 27 significantly overexpressed, and 30 significantly underexpressed, umiRNAs in ccRCC. Among them, they tested the two most overexpressed umiRNAs (miR-122 and miR-15b), plus four more randomly chosen overexpressed miRNAs (miR-1271, miR-629, miR-625, and miR-93), and the most underexpressed miRNA (miR-1260a) plus another randomly chosen underexpressed miRNA (miR-369). The authors compared urinary expression levels in patients versus healthy controls and concluded that the combined use of urinary miR-122, miR-1271, and miR-15b, together with imaging controls, allowed them to diagnose ccRCC with high sensitivity and specificity [40].

In conclusion, one of the open challenges in renal cancer identification is differentiation with benign masses, such as oncocytoma [91]. In this regard, in 2018, von Brandenstein et al. aimed at finding urinary miRNAs that would allow them to discriminate benign and malign masses. Thus, they collected urinary samples from 26 patients with renal masses and 17 urine samples of healthy volunteers or patients with other pathologies. They found that miR-498 (associated with the formation of the oncocytoma-specific slice-form of vimentin, Vim3), miR-183 (associated with increased CO2 levels), miR-205, and miR-31 were specific urinary miRNAs guiding the diagnosis for benign oncocytoma [42]. Accordingly, Di Meo et al. tested the sensibility of mi-RNA in discriminating benign oncocytoma from early-stage ccRCC, identifying miR-432-5p and miR-532-5p as presenting the higher discriminatory power, followed by miR-10a-5p, miR-144-3p, miR-28-3p, miR-326, miR-603, and miR-93-3p. In particular, miR-93-3p was identified as the only miRNA associated with progressive ccRCC when downregulated (*p* = 0.042) and with longer overall survival when upregulated (*p* = 0.016) [39].

#### 3.2.4. The Role of MicroRNA in the Detection of Upper Tract Urothelial Carcinoma

Urothelial carcinomas (UCs) are the sixth most common tumor in developed countries [50]. They can be localized in the lower (bladder and urethra) and/or the upper (pyelo-caliceal cavities and ureter) urinary tract. While BCs account for 90–95% of UCs, upper tract UCs (UTUCs) are uncommon and account for only 5–10% of UCs [92,93].

In this section, we aimed at focusing on the role of miRNA in UTUCs’ detection.

Back in 2011, Yamada et al. evaluated miRNAs’ expression in clinical samples, using specimens from 104 UC patients who underwent cystectomy, between 2003 and 2007, and urine samples from another series of UC patients (BC, renal pelvic, and ureter (UC)) who had undergone cystectomy, TUR-BT or nephrouretectomy, between 2008 and 2010. Moreover, they collected urine samples from 49 healthy volunteers and 25 urine samples from patients with urinary tract infections (UTIs). They tested miR-96, miR-183, and miR-190, which had appeared to be upregulated in a previous study based on urine from UC patients [94]. Urinary concentrations of miR-190 presented no clinically significant difference between patients and controls, whereas miR-96 and miR-183 were significantly higher in UC patients than in controls or UTI samples (*p* < 0.006) [47].

Matsuzaki et al. analyzed, in 2017, urinary samples of 36 patients diagnosed with UC, and 24 controls (defined as without history of UC), and selected five miRNAs that showed a more than 2.5-fold higher expression and *p*-value < 0.1 in the urinary extracellular vesicles of UC patients, compared to those of healthy volunteers. The authors identified miR-155-5p, miR-15a-5p, miR-21-5p, miR-132-3p, and miR-31-5p as all significantly more expressed in the urinary extracellular vesicles of UC patients compared to those of the control (all *p* < 0.0001). Through logistic multivariate analyses, the authors found that miR-21-5p was the most important predictor of UC (AUC = 0.900) and could be a candidate for early diagnosis of UC even in patients with negative urine cytology [46].

## 4. Conclusions

Since the first report in 2010 on the role of umiRNAs in the diagnosis of BC, additional umiRNAs have been tested as diagnostic tools for detecting further genitourinary cancers. Despite the subsequent large number of published studies, no umiRNA, or sets of them, has been univocally recognized as a powerful instrument that can help physicians in the diagnosis, prognosis, and therapeutic management of genitourinary cancers. In this context, the development of brand-new diagnostic tools allowing the early detection of cancers is still evolving in biomedical research. By organizing the current literature related to this topic, we can recommend further research to find the best way to implement umiRNAs in everyday clinical practice.

## Figures and Tables

**Figure 1 ijms-24-10846-f001:**
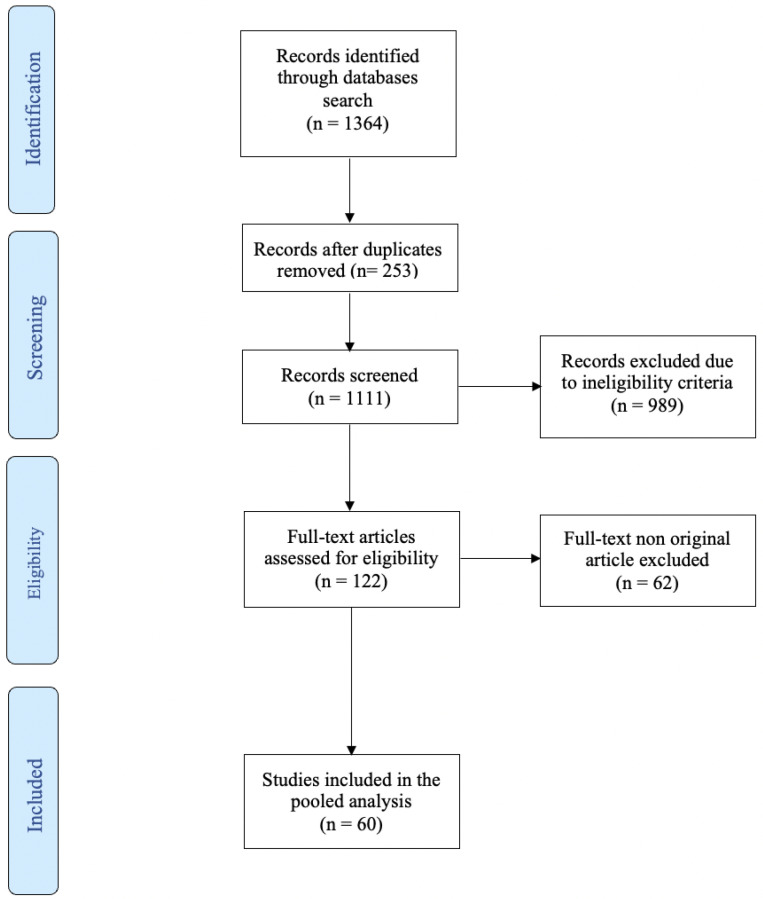
Prisma flow chart—study selection with inclusion and exclusion criteria of reviewed studies.

**Table 1 ijms-24-10846-t001:** Characteristics of the included studies of bladder cancer classified according to year of publication (2023–2012).

Authors	Year of Publication	Number of Patients (BC/Ctl)	Study Design	Target (umiRNA in BC)	Primary Findings	Purpose
Mamdouh et al. [13]	2023	111/25	Retrospective	miR-200 (↑) miR-145 (↑) miR-21 (↑)	Positive correlation (*p* = 0.02) high and low grade > controls (*p* = 0.01) high and low grade > controls (*p* = 0.05) high and low grade > controls	Diagnostic and surveillance
Moisoiu et al. [14]	2022	15/16	Retrospective	Panel of three miRNAs: miR-34a-5p (↑) miR-205-5p (↑) miR-210-3p (↑)	AUC 0.92 (miRNA + SERS)	Diagnostic
Lin et al. [15]	2021	180/100	Retrospective	let-7c-5p (↑) miR-146a-5p (↑) miR-149-5p (↑) miR-193a-5p (↑) miR-423-5p (↑)	Positive correlation BC > Ctl	Diagnostic
Baumgart et al. [16]	2019	37/0	Retrospective	miR-146 (↑)	Positive correlation High grade > low grade	Diagnostic
Braicu et al. [17]	2019	23/23	Retrospective	miR-141-3p (↑) miR-205-5p (↑) miR-139-5p (↓) miR-143-5p (↓) miR-200b-3p (↑)	AUC 0.86 (overall) AUC 0.89 (overall) BC < Ctl BC < Ctl BC > Ctl	Diagnostic
Pardini et al. [18]	2018	66/48	Retrospective	Panel of three miRNAs: let-7c-5p (↑) miR-30a-5p (↑) miR-486-5p (↓)	AUC 0.70 (overall) AUC 0.73 (low-grade NMIBC) AUC 0.95 (high-grade NMIBC) AUC 0.99 (MIBC)	Diagnostic and surveillance
Huang et al. [19]	2018	80/86	Retrospective	miR-20a (↑)	Positive correlation (*p* < 0.001) Associated with larger tumor size and advanced tumor grade in NMIBC (all *p* < 0.05)	Diagnostic and surveillance
Ghorbanmehr et al. [20]	2018	45/20	Retrospective	miR-21-5p (↑) miR141-3p (↑) mir205-5p (↑)	Positive correlation 84% SN, 59% SP; AUC 0.76 (overall) 71% SN, 71% SP; AUC 0.74 (overall) 82% SN, 62% SP; AUC 0.73 (overall)	Diagnostic
Piao et al. [21]	2018	35/20	Retrospective	miR-6124 to miR-4511 ratio (↑)	Positive correlation (AUC: 0.888, 91.5% SN, 76.2% SP) (*p* < 0.001)	Diagnostic
Hofbauer et al. [22]	2018	87/115	Retrospective	Panel of six miRNAs: Let-7c (↓) miR-135a (↓) miR-135b (↑) miR-148a (↓) miR-204 (↓) miR-345 (↑)	AUC 0.88 (overall) AUC 0.91 (MIBC)	Diagnostic
Andreu et al. [23]	2017	36/9	Retrospective	miR-146 (↑)	Low grade > high grade	Diagnostic and surveillance
Sasaki et al. [24]	2016	28/19	Retrospective	miR-146a-5p (↑)	Positive correlation (AUC = 0.773, 95% CI, 0.701–0.892) (*p* = 0.014) (*p* = 0.0436) (high-grade > low-grade) (*p* = 0.1391) (MIBC > NMIBC)	Diagnostic
Zhang et al. [25]	2016	162/162	Retrospective	miR-155 (↑)	Positive correlation (AUC = 0.804; 95% CI, 0.756–0.845, 80.2% SN, 84.6% SP)(NMIBC)	Diagnostic
Kim et al. [26]	2013	138/144	Retrospective	miR-214 (↑)	Positive correlation 20.08 ± 3.21 vs. 18.96 ± 2.68, (*p* = 0.002) (NMIBC)	Diagnostic
Mengual et al. [27]	2013	181/136	Retrospective	Panel of six miRNAs: miR-18a (↑) miR-25 (↑) miR-140-5p (↓) miR-187 (↑) miR-142-3p (↓) miR-204 (↓)	84.8% SN, 86.5% SP; AUC 0.92 (overall) 87.1% SN, 86.5% SP (MIBC)	Diagnostic and surveillance
Yun et al. [28]	2012	207/144	Retrospective	miR-145 (↓) miR-200a (↓)	Negative correlation miR-145 (AUC = 0.729; 77.8% SN, 61.1% SP) (NMIBC < healthy controls) miR-145 (AUC = 0.79; 84.1% SN, 61.1% SP) (MIBC < healthy controls) miR-145 (*p* = 0.036) (MIBC < NMIBC) miR-200a (*p* < 0.001) (MIBC and NMIBC < healthy controls)	Diagnostic and surveillance
Hanke et al. [29]	2010	29/18	Retrospective	miR-126 (↑) miR-182 (↑) miR-199a (↑) miR-126 to miR-152 ratio (↑)	Positive correlation miR-126 to miR-152 ratio (AUC = 0.768; 72% SN, 82% SP)	Diagnostic

**Abbreviations:** BC: bladder cancer; Ctl: control participants; umiRNA: urinary microRNA; NMIBC: non-muscle-invasive bladder cancer; AUC: area under the curve; CI: confidence interval; *p*: *p*-value; SN: sensitivity; SP: specificity; SERS: surface-enhanced Raman spectroscopy. Arrows indicates higher or lower levels of related umiRNAs in related studies.

**Table 2 ijms-24-10846-t002:** Characteristics of the included studies of prostate cancer classified according to year of publication (2022–2015).

Authors	Year of Publication	Number of Patients (PCa/Ctl)	Study Design	Target (umiRNA in PCa)	Primary Findings	Purpose
Lee et al. [30]	2022	6/8	Retrospective	miR-21-5p, miR-574-3p, and miR6880-5p (↑)	Positive correlation in CRPC miR-21-5p, miR-574-3p (*p* < 0.05) miR6880-5p (*p* < 0.01)	Surveillance
Kang et al. [31]	2022	63/53	Retrospective	miR-H9 to miR-3659 ratio (↑)	Positive correlation (AUC = 0.803, 95% CI) (*p* = 0.001)	Diagnostic
Byun et al. [32]	2021	14/5	Retrospective	miR-1913 to miR-3659 ratio (↑)	Positive correlation (AUC = 0.7, 95% CI, 61.4% SN, 71.8% SP)	Diagnostic
Hasanoglu et al. [33]	2021	8/30	Retrospective	miR-320a (↑)	Positive correlation *p* = 0.0168	Diagnostic
Markert et al. [34]	2021	28/25	Retrospective	miR-19b and miR-26a (↓)	Negative correlation AUC = 0.7	Diagnostic
Ghorbanmehr et al. [20]	2020	23/42	Retrospective	miR-21-5p (↑) mi-R-141-3p (↑) miR-205p (↑)	Positive correlation *p* = 0.001 *p* = 0.005 *p* = 0.020	Diagnostic
Foj et al. [35]	2017	60/10	Retrospective	miR-21, miR-141, and miR-375 (↑) let-7c	Positive correlation miR-21 (*p* = 0.001) miR-141(*p* = 0.033); higher Gleason score (*p* = 0.034) miR-375 (*p* = 0.038) let-7c (no correlation)	Diagnostic
Rodriguez et al. [36]	2017	28/19	Retrospective	miR-196a-5p and miR-501-3p (↓)	Negative correlation miR-196a-5p (AUC = 0.73, 95% CI 0.56 to 0.86) miR-501-3p (AUC = 0.69%, 95% CI 0.52 to 0.85)	Diagnostic
Salido-Guadarrama et al. [37]	2016	73/70	Retrospective	miR-100 and miR-200b (↑)	Positive correlation (*p* = 0.0355; Spearman coefficient = 0.18)	Diagnostic
Stephan et al. [38]	2015	38/38	Retrospective	miR-183 and miR-205	No correlation	Diagnostic

**Abbreviations:** PCa: prostate cancer; Ctl: control participants; umiRNA: urinary microRNA; AUC: area under the curve; CI: confidence interval; *p*: *p*-value; SN: sensitivity; SP: specificity; CRPC: castration-resistant prostate cancer. Arrows indicates higher or lower levels of related umiRNAs in related studies.

**Table 3 ijms-24-10846-t003:** Characteristics of the included studies of renal cancer classified according to year of publication (2020–2012).

Authors	Year of Publication	Number of Patients (RCC/Ctl)	Study Design	Target (umiRNA in RCC)	Primary Findings	Purpose
Di Meo et al. [39]	2020	6/8	Retrospective	miR-432-5p and miR-532-5p (↑↑) miR-10a-5p, miR-144-3p, miR-28-3p, miR-326, miR-328-3p, miR-603, and miR-93-3p (↑)	Positive correlation miR-432-5p (AUC: 0.71, 95% CI: 0.59 to 0.83, *p* = 0.003) miR-532-5p (AUC: 0.70, 95%CI: 0.57–0.82, *p* = 0.007) miR-10a-5p (AUC: 0.66, 95% CI: 0.53–0.79) miR-144-3p (AUC: 0.68, 95% CI: 0.55–0.81) miR-28-3p (AUC: 0.65, 95% CI: 0.52–0.78) miR-326 (AUC: 0.68, 95% CI: 0.55–0.81) miR-328-3p (AUC: 0.65, 95% CI: 0.52–0.78) miR-603 (AUC: 0.67, 95% CI: 0.55–0.80), and miR-93-3p (AUC: 0.68, 95% CI: 0.54–0.81), all *p* < 0.05	Diagnostic
Cochetti et al. [40]	2020	13/14	Retrospective	Panel of: miR-122, miR-1271, miR-15b (↑)	(100% SN (95% CI 75–100%), and 86% SP (95% CI 57–98%), AUC of 0.96 and *p* < 0.001)	Diagnostic
Song et al. [41]	2019	70/30	Retrospective	miR-30c-5p (↓)	Negative correlation (68.57% SN and 100%SP)	Diagnostic
von Brandenstein et al. [42]	2018	26/17	Retrospective	miR-498, miR-183, miR-205, and miR-31(↑)	Positive correlation with oncocytoma	Diagnostic
Mytsyk et al. [43]	2018	67/15	Retrospective	miR-15a (↑)	Positive correlation between miR-15a levels and tumor size (98.1% SP, 100% SN, AUC = 0.955, *p* < 0.001)	Diagnostic
Li et al. [44]	2017	75/45	Retrospective	miR-210 (↑)	Positive correlation *p* < 0.001 (SN of 57.8% and SP of 80.0%)	Diagnostic
Fedorko et al. [45]	2017	69/36	Retrospective	all let-7 miRNAs (let-7a, let-7b, let-7c, let-7d, let-7e and let-7g (↑)	Positive correlation (AUC = 0.8307, 71% SN, 81% SP), all *p* < 0.05.	Diagnostic
von Brandenstein et al. [42]	2012	25/5	Retrospective	miR-15a (↑)	Positive correlation (*p* not reported)	Diagnostic

**Abbreviations:** RCC: renal cell carcinoma; Ctl: control participants; umiRNA: urinary microRNA; AUC: area under the curve; CI: confidence interval; *p*: *p*-value; SN: sensitivity; SP: specificity. Arrows indicates higher or lower levels of related umiRNAs in related studies.

**Table 4 ijms-24-10846-t004:** Characteristics of the included studies of upper tract urothelial carcinoma classified according to year of publication (2017–2011).

Authors	Year of Publication	Number of Patients (UTUC/Ctl)	Study Design	Target (umiRNA in UTUC)	Primary Findings	Purpose
Matsuzaki et al. [46]	2017	36/26	Retrospective	miR-155-5p, miR-15a-5p, miR-21-5p, miR-132-3p and miR-31-5p (↑)	Positive correlation in UTUC (all *p* < 0.001) miR-21-5p (AUC = 0.900)	Diagnostic
Yamada et al. [47]	2011	<104/74	Retrospective	miR-190 (=) miR-96 and miR-183 (↑)	Positive correlation (*p* = 0.006)	Diagnostic

**Abbreviations:** UTUC: upper tract urothelial carcinoma; Ctl: control participants; umiRNA: urinary microRNA; AUC: area under the curve; CI: confidence interval; *p*: *p*-value; SN: sensitivity; SP: specificity. Arrows indicates higher or lower levels of related umiRNAs in related studies.

## Data Availability

Data are available in scientific databases.

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
