# Peer review of "Urinary MicroRNAs as Biomarkers of Urological Cancers: A Systematic Review"

_ijms, 2023, doi:10.3390/ijms241310846_

Round 1

Reviewer 1 Report

Identification of new and novel biomarkers is a significant ongoing approach. This will lead to a foolproof diagnosis of the diseases, and early detection can lead to proper intervention. Here microRNAs were searched in the literature to identify potential and significant molecules that can potentially become new biomarkers for new or existing diseases. The data from the literature is mined and removed from all duplications, and a trend in the data was postulated. Table 1-4 is a collective representation of the study with the sample size and correlation to the markers. There is a lot of interest in this kind of study as diagnostic companies are looking for at-home kits or simple test kits, and this study will help them move forward in creating many more of such easy, simple tests that can detect cancer early.

No significant revisions are required.  

Author Response

Identification of new and novel biomarkers is a significant ongoing approach. This will lead to a fool-proof diagnosis of the diseases, and early detection can lead to proper intervention. Here microRNAs were searched in the literature to identify potential and significant molecules that can potentially become new biomarkers for new or existing diseases. The data from the literature is mined and removed from all duplications, and a trend in the data was postulated. Table 1-4 is a collective representation of the study with the sample size and correlation to the markers. There is a lot of interest in this kind of study as diagnostic companies are looking for at-home kits or simple test kits, and this study will help them move forward in creating many more of such easy, simple tests that can detect cancer early.

No significant revisions are required.

A1. We thank the Reviewer 1 for the overall comment on our manuscript, we greatly appreciate.

Reviewer 2 Report

To the authors of the manuscript: “Urinary micro-RNAs as biomarkers of urological cancers: a systematic review”

It is true that the topic chosen for the systematic review is highlighted, since in recent years there has been a growing interest in the search for and identification of biomarkers through less invasive techniques that ensure better management of the disease and improve the quality of patient health, especially with regard to genitourinary cancers, which are among the most expensive for health systems and are based on invasive techniques for monitoring patients every few months and for years, thus worsening both their mental and physical health.

However, after carefully reading the manuscript, I find that the review is scarce, although it is based on some current references (less than 10 years old), although many of them are not very relevant. I think it is important to introduce the concept of liquid biopsy well and compare it with other standard detection systems so that the reader understands the problem as well as the object of this study. There is extensive literature on these methods of detection and monitoring of patients. For example:

G. Mowatt, S. Zhu, M. Kilonzo, et al. Systematic review of the clinical effectiveness and cost-effectiveness of photodynamic diagnosis and urine biomarkers (FISH, ImmunoCyt, NMP22) and cytology for the detection and follow-up of bladder cancer.Health Technol Assess, 14 (2010), pp. 1-331, 10.3310/hta14040

There is controversy between line 40 (where it is commented that miRNAS are downregulated in prostate cancer) and lines 225-226 (where miR-183 is exemplified, which is upregulated). It must be taken into account that there is not usually a single microRNA as a biomarker, but rather a set of them, both up and down-regulated, and that this is what generates a signature that can be used as a tool, sometimes in diagnosis. , others in detection, others in follow-up, others in recurrence and even in progression. In addition, these signatures tend to have better specificity and sensitivity, as well as allowing us to understand the mechanism and behavior of the tumor, either in the different subtypes or in its different stages. I think that is something that should also be addressed in this review.

It is also important to discern well why the object of study is urine and what problems may exist in serum/plasma. In addition to microRNAs and not other molecules such as peptides, CTCs, ctDNA, exoxomes, vesicles or metabolites, which in the case of bladder and prostate tumors are having great acceptance and there are already validations by magnetic resonance. I propose some reference that I think is important and should be included in the manuscript:

M. Martínez-Fernández, J.M. Paramio, M. Dueñas.RNA detection in urine: from RNA extraction to good normalizer molecules. J Mol Diagn, 18 (2016), pp. 15-22, 10.1016/j.jmoldx.2015.07.008

Cristian Suarez-Cabrera et al. BlaDimiR: A Urine-based miRNA Score for Accurate Bladder Cancer Diagnosis and Follow-up. European Urology,Volume 82, Issue 6,2022,Pages 663-667, https://doi.org/10.1016/j.eururo.2022.08.011.

The tables are chaotic. I am much more interested in seeing its applicability (if it is for diagnosis, prognosis, response to treatment... etc) for what type of tumor it is (ex. NMIBC or MIBC or both), not as "first discoveries". One column for each term: specificity/sensitivity/AUC, another for its applicability and tumor type…

 In the body of the manuscript, they dont make clear an idea that connects the chosen studies, they only lists and summarizes the studies in a few lines. For this, there are already tables that summarize it.

The workflow of the study has no interest beyond describing it in materials and methods. After the screening criteria they have chosen, it is more relevant to choose those tests that are validated or in the process of being validated, since what is sought is real biomarkers that can be used in the clinic. Here is a reference to some examples:

M.J. Ribal, L. Mengual, J.J. Lozano, et al. Gene expression test for the non-invasive diagnosis of bladder cancer: a prospective, blinded, international and multicenter validation study. Eur J Cancer, 54 (2016), pp. 131-138, 10.1016/j.ejca.2015.11.003

M. Hanke, K. Hoefig, H. Merz, et al. A robust methodology to study urine microRNA as tumor marker: microRNA-126 and microRNA-182 are related to urinary bladder cancer. Urol Oncol, 28 (2010), pp. 655-661, 10.1016/j.urolonc.2009.01.027

I recommend to the authors a reference that I think is important and that can be of great help when rewriting the manuscript and generating the tables.

I. Lodewijk, M. Dueñas, C. Rubio, et al.Liquid biopsy biomarkers in bladder cancer: a current need for patient diagnosis and monitoring. Int J Mol Sci, 19 (2018), p. 2514, 10.3390/ijms19092514

Minor comments:

• Reference should always be made to genitourinary and not urinary cancers, since the study includes the prostate.

• Sometimes they use different acronyms to define the same thing (BCa /BC)

• In all the sections they use the words "in conclusion" instead of "finally". A conclusion is a final idea that encompasses everything previously proposed, when in fact what they describe is a final study. Please write properly.

• wrong referenced literature: nº 10, 42, 47, 82

• Literature without year of publication: nº 21

• Supplementary: it is the same workflow of the manuscript.

Author Response

Reviewer 2

To the authors of the manuscript: “Urinary micro-RNAs as biomarkers of urological cancers: a systematic review”

It is true that the topic chosen for the systematic review is highlighted, since in recent years there has been a growing interest in the search for and identification of biomarkers through less invasive techniques that ensure better management of the disease and improve the quality of patient health, especially with regard to genitourinary cancers, which are among the most expensive for health systems and are based on invasive techniques for monitoring patients every few months and for years, thus worsening both their mental and physical health.

However,

Q1. after carefully reading the manuscript, I find that the review is scarce, although it is based on some current references (less than 10 years old), although many of them are not very relevant.

A1. We thank Reviewer 2 for this comment. As reported in the manuscript, we evaluated all the original articles published in the current literature without any temporal criteria.

Q2. I think it is important to introduce the concept of liquid biopsy well and compare it with other standard detection systems so that the reader understands the problem as well as the object of this study. There is extensive literature on these methods of detection and monitoring of patients. For example:

  1. Mowatt, S. Zhu, M. Kilonzo, et al. Systematic review of the clinical effectiveness and cost-effectiveness of photodynamic diagnosis and urine biomarkers (FISH, ImmunoCyt, NMP22) and cytology for the detection and follow-up of bladder cancer.Health Technol Assess, 14 (2010), pp. 1-331, 10.3310/hta14040

A2. We thank Reviewer 2 for this important comment. We briefly discussed liquid biopsy in the “Introduction” section.

Q3. There is controversy between line 40 (where it is commented that miRNAS are downregulated in prostate cancer) and lines 225-226 (where miR-183 is exemplified, which is upregulated). It must be taken into account that there is not usually a single microRNA as a biomarker, but rather a set of them, both up and down-regulated, and that this is what generates a signature that can be used as a tool, sometimes in diagnosis. , others in detection, others in follow-up, others in recurrence and even in progression. In addition, these signatures tend to have better specificity and sensitivity, as well as allowing us to understand the mechanism and behavior of the tumor, either in the different subtypes or in its different stages. I think that is something that should also be addressed in this review.

A3. We thank Reviewer 2 for this pertinent comment. We corrected the abstract in the line indicated.

As declared in the manuscript, we considered all the original articles published regarding the role of umi-RNA in genitourinary cancers. Some of these focused on single mi-RNA and not on the set of them, thus we reported the results as it is. When included studies analysed sets of mi-RNAs, we reported the results accordingly.

Q4. It is also important to discern well why the object of study is urine and what problems may exist in serum/plasma. In addition to microRNAs and not other molecules such as peptides, CTCs, ctDNA, exoxomes, vesicles or metabolites, which in the case of bladder and prostate tumors are having great acceptance and there are already validations by magnetic resonance. I propose some reference that I think is important and should be included in the manuscript:

  1. Martínez-Fernández, J.M. Paramio, M. Dueñas.RNA detection in urine: from RNA extraction to good normalizer molecules. J Mol Diagn, 18 (2016), pp. 15-22, 10.1016/j.jmoldx.2015.07.008

Cristian Suarez-Cabrera et al. BlaDimiR: A Urine-based miRNA Score for Accurate Bladder Cancer Diagnosis and Follow-up. European Urology,Volume 82, Issue 6,2022,Pages 663-667, https://doi.org/10.1016/j.eururo.2022.08.011.

 A4. We thank Reviewer 2 for the insightful comment. We implemented the “Introduction” section accordingly.

Q5. The tables are chaotic. I am much more interested in seeing its applicability (if it is for diagnosis, prognosis, response to treatment... etc) for what type of tumor it is (ex. NMIBC or MIBC or both), not as "first discoveries". One column for each term: specificity/sensitivity/AUC, another for its applicability and tumor type…

A5. We thank Reviewer 2 for this comment. We reported in “Table 1” the applicability of umi-RNAs as reported in the selected studies as requested.

Q6. In the body of the manuscript, they dont make clear an idea that connects the chosen studies, they only lists and summarizes the studies in a few lines. For this, there are already tables that summarize it.

The workflow of the study has no interest beyond describing it in materials and methods. After the screening criteria they have chosen, it is more relevant to choose those tests that are validated or in the process of being validated, since what is sought is real biomarkers that can be used in the clinic. Here is a reference to some examples:

M.J. Ribal, L. Mengual, J.J. Lozano, et al. Gene expression test for the non-invasive diagnosis of bladder cancer: a prospective, blinded, international and multicenter validation study. Eur J Cancer, 54 (2016), pp. 131-138, 10.1016/j.ejca.2015.11.003

  1. Hanke, K. Hoefig, H. Merz, et al. A robust methodology to study urine microRNA as tumor marker: microRNA-126 and microRNA-182 are related to urinary bladder cancer. Urol Oncol, 28 (2010), pp. 655-661, 10.1016/j.urolonc.2009.01.027

A6. We thank Reviewer 2 for this comment. We chose the temporal criteria to report the results of our research. This criterion has been chosen to highlight and summarise improvements made during the years from the scientifical community in this field. Further validation of these biomarkers in the real-life scenario is needed. We modified the aim of the study in the “Introduction” and “Conclusions” sections accordingly.

Q7. I recommend to the authors a reference that I think is important and that can be of great help when rewriting the manuscript and generating the tables.

  1. Lodewijk, M. Dueñas, C. Rubio, et al.Liquid biopsy biomarkers in bladder cancer: a current need for patient diagnosis and monitoring. Int J Mol Sci, 19 (2018), p. 2514, 10.3390/ijms19092514

Q7. We thank Reviewer 2 for the pertinent suggestion. We implemented our manuscript with considerations regarding literature findings, highlighting the utility of these.

A8. Minor comments:

  • Reference should always be made to genitourinary and not urinary cancers, since the study includes the prostate.
  • Sometimes they use different acronyms to define the same thing (BCa /BC)
  • In all the sections they use the words "in conclusion" instead of "finally". A conclusion is a final idea that encompasses everything previously proposed, when in fact what they describe is a final study. Please write properly.
  • wrong referenced literature: nº 10, 42, 47, 82
  • Literature without the year of publication: nº 21
  • Supplementary: it is the same workflow of the manuscript.

Q8. We thank Reviewer 2 for all the minor comments. We addressed all the required improvements in the main text and references.

Round 2

Reviewer 2 Report

I am grateful to the authors who have taken the trouble to try to cover the recommendations given so that the manuscript can be published as well as including a cover letter. I leave some changes still necessary

·         Line 84-88:

Redundance, rewrite the paragraph

In consequence, considering the high worldwide prevalence of genitourinary cancers and the growing interest in the role of miRNAs as non-invasive diagnostic biomarkers for gen-itourinary cancers, the current systematic review aims at summarizing the role of umiRNAs in any genitourinary cancer in order to lay the foundation for further validation studies.

·         Line 115;121:

(umiRNAs, umRNAs). Check the abbreviations in all manuscript

·         line 136-131:

Include the reference to Kim et al at the end of the paragraph

·         Table 1. Characteristics of the included studies of bladder cancer classified according to year of publication (2023-2012)

4th column: Target (microRNA in PCa). Change  BC, not PCa

·         Include the applicability/the purpose in the rest of the tables (CPa and CCR), it is only in BC table.

Author Response

we sincerely thank the Reviewers for helping us in improving the overall quality of the manuscript.
